# SARS-CoV-2 Antibody Response against Mild-to-Moderate Breakthrough COVID-19 in Home Isolation Setting in Thailand

**DOI:** 10.3390/vaccines10071131

**Published:** 2022-07-15

**Authors:** Pichanun Mongkolsucharitkul, Apinya Surawit, Sureeporn Pumeiam, Nitat Sookrung, Anchalee Tungtrongchitr, Pochamana Phisalprapa, Naruemit Sayabovorn, Weerachai Srivanichakorn, Chaiwat Washirasaksiri, Chonticha Auesomwang, Tullaya Sitasuwan, Thanet Chaisathaphol, Rungsima Tinmanee, Methee Chayakulkeeree, Pakpoom Phoompoung, Watip Tangjittipokin, Sansnee Senawong, Gornmigar Sanpawitayakul, Saipin Muangman, Korapat Mayurasakorn

**Affiliations:** 1Siriraj Population Health and Nutrition Research Group, Department of Research Group and Research Network, Faculty of Medicine Siriraj Hospital, Mahidol University, Bangkok 10700, Thailand; pichanun.mon@mahidol.ac.th (P.M.); apinya.sua@mahidol.ac.th (A.S.); sureeporn.pum@mahidol.ac.th (S.P.); 2Center of Research Excellence on Therapeutic Proteins and Antibody Engineering, Department of Parasitology, Faculty of Medicine Siriraj Hospital, Mahidol University, Bangkok 10700, Thailand; nitat.soo@mahidol.ac.th (N.S.); anchalee.tun@mahidol.ac.th (A.T.); 3Division of Ambulatory Medicine, Department of Medicine, Faculty of Medicine Siriraj Hospital, Mahidol University, Bangkok 10700, Thailand; pochamana.phi@mahidol.ac.th (P.P.); naruemit.say@mahidol.ac.th (N.S.); weerachai.srv@mahidol.ac.th (W.S.); chaiwat.was@mahidol.ac.th (C.W.); chonticha.aue@mahidol.ac.th (C.A.); tullaya.rua@mahidol.ac.th (T.S.); thanet.chs@mahidol.ac.th (T.C.); rungsima.tin@mahidol.ac.th (R.T.); 4Division of Infectious Diseases and Tropical Medicine, Department of Medicine, Faculty of Medicine Siriraj Hospital, Mahidol University, Bangkok 10700, Thailand; methee.cha@mahidol.ac.th (M.C.); pakpoom.pho@mahidol.ac.th (P.P.); 5Department of Immunology, Faculty of Medicine Siriraj Hospital, Mahidol University, Bangkok 10700, Thailand; watip.tan@mahidol.ac.th (W.T.); sansnee.sen@mahidol.ac.th (S.S.); 6Division of Ambulatory Paediatrics, Department of Paediatrics, Faculty of Medicine Siriraj Hospital, Mahidol University, Bangkok 10700, Thailand; gornmigar.win@mahidol.ac.th; 7Department of Anesthesiology, Faculty of Medicine Siriraj Hospital, Mahidol University, Bangkok 10700, Thailand; saipin.mun@mahidol.ac.th

**Keywords:** breakthrough infection, neutralizing antibody, Delta, Omicron, COVID-19, immunogenicity

## Abstract

Background: In December 2021, Omicron replaced Delta as the dominant coronavirus disease 2019 (COVID-19) variant in Thailand. Both variants embody diverse epidemiological trends and immunogenicity. We investigated whether Delta and Omicron patients’ biological and clinical characteristics and immunogenicity differed post-COVID-19 infection. Methods: This retrospective cohort study investigated the clinical outcomes and laboratory data of 5181 patients with mild-to-moderate COVID-19 (Delta, 2704; Omicron, 2477) under home isolation. We evaluated anti-receptor-binding domain immunoglobulin G (anti-RBD IgG) and surrogate viral neutralizing (sVNT) activity in 495 individuals post-COVID-19 infection during the Delta pandemic. Results: Approximately 84% of all patients received favipiravir. The median cycle threshold (Ct) values were lower for Omicron patients than Delta patients (19 vs. 21; *p* < 0.001), regardless of vaccination status. Upper respiratory tract symptoms were more frequent with Omicron patients than Delta patients. There were no significant associations between Ct and Omicron symptoms (95% confidence interval 0.98–1.02). A two-dose vaccine regimen reduced hospital readmission by 10% to 30% and death by under 1%. Anti-RBD IgG and sVNT against Delta were higher among older individuals post-COVID-19 infection. Older individuals expressed anti-RBD IgG and sVNT for a more extended period after two-dose vaccination than other age groups. Conclusions: After a full vaccination course, breakthrough mild-to-moderate Delta and Omicron infections have limited immunogenicity. Prior infections exert reduced protection against later reinfection or infection from novel variants. However, this protection may be sufficient to prevent hospitalization and death, particularly in countries where vaccine supplies are limited.

## 1. Introduction

In 2021, Thailand faced multiple rapid waves of coronavirus disease 2019 (COVID-19). Social distancing and vaccines were encouraged as survival tools for people to circumvent the threats [1,2]. CoronaVac (a whole-cell inactivated vaccine, Sinovac, Life Science) and ChAdOx1 (a modified chimpanzee DNA adenovirus-vectored vaccine, AstraZeneca/Oxford) were more widely used than other vaccines by Thais [3]. The CoronaVac vaccine is based on a form of severe acute respiratory syndrome coronavirus 2 (SARS-CoV-2) that has been weakened and safely generates an immune response [4]. The AstraZeneca vaccine is based on the virus’s genetic instructions containing the SARS-CoV-2 structural surface glycoprotein antigen (spike protein; nCoV-19) gene to build the spike protein. The spike protein fragments can then be recognized by the immune system [4,5]. Both vaccines were efficacious against symptomatic COVID-19 caused by the Wuhan strain, but they proved to be less effective against other COVID-19 variants of concern, including Delta and Omicron [6,7]. The vaccine effectiveness is low and wanes faster against infection and mild-to-moderate symptomatic disease but is high against severe disease caused by the Omicron variant. Evidence indicated that the vaccine effectiveness against severe disease outcome after receipt of a primary series with either CoronaVac or AstraZeneca or a booster dose increased to >70% for all vaccines within the first 3 months after a final dose [8]. Omicron displaced Delta as the predominant variant during the study period [1]. Randomly selected SARS-CoV-2 variants captured by surveillance conducted by the Department of Medical Science [6] and worldwide during weeks 4 to 10 of 2022 demonstrated that almost all new infections in Thailand were due to Omicron (99.6%) (Appendix A).

Since 2020, the Thai National Treatment Guidelines for COVID-19 from the Ministry of Public Health [9] recommended that favipiravir, a broad-spectrum nucleotide analog targeting the viral RNA-dependent RNA polymerase [10], be the treatment option for patients at increased risk of severe disease and mild severity of pneumonia. It has widely been repurposed to treat mild-to-moderate cases of COVID-19, including Delta and Omicron. In our experiences and in earlier studies, it showed promising results in patients with mild-to-moderate COVID-19 with well-tolerated side effects [11,12]. Remdesivir [13], a monophosphoramidate prodrug of the nucleoside GS-441524, is only recommended for use in severe disease due to limited access. In addition, monipiravir [13], the oral prodrug of beta-D-N4-hydroxycytidine (NHC), and anti-SARS-CoV-2 monoclonal antibodies were not available during the study period.

A combination of waning vaccine-derived immunity and the arrival of the SARS-CoV-2 variants, Delta (B.1.617.2) and Omicron (B.1.1.529), led to breakthrough infections after COVID-19 vaccination or prior infection [7,14]. This greatly overloaded the nation’s public health system and exacerbated socioeconomic disparities [2,15]. In response, home isolation (HI) was implemented for patients with mild-to-moderate symptoms nationwide to combat the overwhelming demand for hospital beds (Appendix A).

The Omicron variant caused less severe disease than other variants [16]. Nevertheless, there were serious concerns about its increased transmissibility [17], potential for reduced sensitivity to neutralizing antibodies, and newly emerged lineages (BA.4 and BA.5) [18,19,20]. Very few studies have verified the guidance for vaccination after a mild-to-moderate COVID-19 infection, particularly in countries where the provision of full vaccination courses is challenging [21]. Here, we present the results of our analyses of retrospective information from mildly to moderately symptomatic individuals who were seropositive for SARS-CoV-2. All patients were in the HI system during the Delta and Omicron pandemics. They were treated between July 2021 and March 2022.

## 2. Materials and Methods

This retrospective cohort study evaluated the treatment outcomes and immunogenicity of mild-to-moderate COVID-19 patients admitted to the HI system of Siriraj Hospital, Bangkok, Thailand. The Institutional Review Board (IRB), Faculty of Medicine Siriraj Hospital, Mahidol University reviewed and approved the follow-up study (COA: Si 732/2021 and Si 833/2021), and it was registered in ClinicalTrials.gov (NCT05328479).

### 2.1. Study Population

The data collection reported here was performed between 8 July 2021 and 15 March 2022. The study population comprised 2704 and 2477 patients during the Delta (before November 2021) and Omicron (after 12 January 2022) pandemics. All had been positive with SARS-CoV-2, determined via reverse transcriptase–polymerase chain reaction (RT-PCR) testing. Someone who met the inclusion criteria was considered to have mild symptoms, or perhaps be asymptomatic, and was referred to the Siriraj-Home system (SI-Home) in which medicine is delivered by health personnel within 24 h rather than being relegated to a field hospital or another potentially unpleasant arrangement. Data relating to clinical information and laboratory test findings were retrieved (after IRB approval) from patients’ electronic medical records without any personal identifiable information. The study protocol and guidelines for COVID-19 standard care were based on national and World Health Organization recommendations [15,22]. 

### 2.2. Patient Selection and Procedures

A subset of 495 patients (age ≥ 12 years) were recruited for a reactogenicity and immunogenicity follow-up study after COVID-19 recovery at 21 to 150 days post-COVID-19 onset. All participants provided informed consent for this study. They were tested for SARS-CoV-2 antibodies and a surrogate virus neutralization (sVNT) against SARS-CoV-2 Wuhan and Delta variants (Figure 1). The patients were classified into different exposure groups based on vaccination status prior to COVID-19 infection, study antibody, and PCR test (Appendix A). 

### 2.3. Outcome Measures

The rationale for mild-to-moderate COVID-19 treatment is described in Appendix A. In brief, the primary treatment strategy in Thailand included early favipiravir treatment and recommended outpatient antiviral therapies. The primary outcome was a comparison of patients’ baseline clinical and biological characteristics with Delta and Omicron variants of SARS-CoV-2 infections in the HI system. Treatment groups were categorized into 3 groups: (1) symptomatic treatment (S), (2) symptomatic treatment plus favipiravir treatment (Favi), and (3) symptomatic treatment plus favipiravir and dexamethasone treatment (Favi/Dexa). The “date of disease onset” was defined as the day when new-onset, self-reported respiratory symptoms were observed. The durations from illness onset to first hospital admission, first favipiravir treatment, and discharge up to 14 days were measured. Viral loads were considered in cycle threshold (Ct) value analyses. Analyses considered viral loads for comparisons of Ct values by the vaccine exposure groups and self-reported symptoms. A Ct value ≥30 corresponded to a copy number threshold <10^6^/mL or less, indicating low viral RNA [23]. 

### 2.4. Diagnosis of COVID-19

The diagnosis of COVID-19 is made based on the detection of ≥2 SARS-CoV-2 genes via RT-PCR from a nasopharyngeal (NP) swab, throat swab, and/or any respiratory samples, as previously described [24]. Our COVID-19 diagnostic assay was a probe-based qualitative RT-PCR probe. The Allplex™ 2019-nCoV Assay (Seegene, Seoul, South Korea) was used for SARS-CoV2 detection. The targeted COVID-19 genes detected here included the nucleocapsid (N), envelope (E) of Sarbecovirus, and RNA-dependent RNA polymerase (RdRp) of COVID-19 according to the manufacturer’s instructions and as described previously [25].

### 2.5. Serological Assays 

Patients were randomly invited to test for anti-SARS-CoV-2 receptor-binding domain immunoglobulin G (anti-RBD IgG, (S1 subunit, No. 06S60)) and SARS-CoV-2 nucleocapsid protein (SARS-CoV-2 IgG II Quant for use with ARCHITECT; Abbott Laboratories, Chicago, IL, USA) [15]. The anti-SARS-CoV-2 RBD IgG assay linearly measures the level of antibody between 21.0 and 40,000.0 arbitrary units (AUs)/mL, which was converted later to the WHO International Standard concentration as binding antibody unit per mL (BAU/mL) following the equation provided by the manufacturer (BAU/mL = 0.142 × AU/mL) [26]. A level greater or equal to the cutoff value of 50 AU/mL or 7.1 BAU/mL was defined as seropositive. A Surrogate Virus Neutralization Test (sVNT) was undertaken against the original (Wuhan) strain and the Delta (B1.1617.2) strain due to its availability during the study period. Briefly, plasma was pre-incubated with horseradish-peroxidase-conjugated receptor-binding domain protein (HRP-conjugated RBD protein). Subsequently, the mixture was transferred to each well containing Streptavidin bound with Biotin-conjugated angiotensin-converting enzyme 2 (ACE2). The plate was washed, and the substrate and stop solution were added. Finally, the optical density absorbance was measured using a spectrophotometer at 450 nm. The inhibition rate was calculated through this formula: $$\mathrm{Inhibition}\;\mathrm{rate}\;\left(\%\right)=\left(1-\frac{{\mathrm{OD}}_{450}\;\mathrm{of}\;\mathrm{Sample}}{{\;\mathrm{OD}}_{450}\;\mathrm{of}\;\mathrm{Negative}\;\mathrm{control}\;}\right)\times 100.$$

Sample diluent was used as the negative control. White blood cell count, C-reactive protein, and D-dimer results were retrieved from electronic medical records from patients who were readmitted to the hospital. 

### 2.6. Statistical Analysis

Multivariable analysis was performed via binary logistic regression for vaccination variables. We used negative binomial mixed models to analyze factors associated with numeric variables, including symptoms and the Chalder fatigue scale [27] (Appendix A). Cox regression analysis was used to analyze the factors of the negative conversion time (NCT) of SARS-CoV-2 RNA. The NCT is closely related to clinical manifestation and disease progression in COVID-19 patients. First, univariate analysis was performed, and the indicators with statistical significance were analyzed with Kaplan–Meier survival analysis. A Cox proportional hazard model was used for multivariate analysis. Normally distributed continuous variables were summarized as the mean ± SD; otherwise, the median (interquartile range, IQR) was used. Categorical variables were expressed using numbers and percentages. The statistical significance of Ct values, IgG, sVNT, and others was determined using Kruskal–Wallis and Dunn’s multiple comparisons tests using GraphPad Prism 9 (GraphPad Software, San Diego, CA, USA) and STATA version 17 (Stata Corp., College Station, TX, USA).

## 3. Results

### 3.1. Demographic and Clinical Data

During the Delta and Omicron pandemics, 2704 and 2477 patients were enrolled, respectively (Table 1). The mean age of the Omicron patients was younger than that of the Delta patients (31.3 ± 12.3 vs. 33.8 ± 11.6 years; *p* < 0.001). The proportion of COVID-19 infections was highest in the group aged 25 years or more during the Delta wave (1470 (54.4%)) and during the Omicron wave (1220, (49.3%)), whereas an increased proportion of COVID-19 infections was observed in the young during the Omicron pandemic (1015, (41%)). The frequent initial symptoms in the Delta wave were low-grade fever (95.2%) and cough (60.7%). With the Omicron wave, however, cough (47.7%) and being asymptomatic (39.1%) were frequently found.

The median duration from disease onset to HI admission was 5.1 days (interquartile range (IQR) = 2.4) for the Delta wave and 2.8 days (IQR = 1.6) for the Omicron wave (*p* = 0.021). The median peak viral RNA based on Ct values during the Omicron wave (19.0 (IQR = 5.7)) was lower than for the Delta wave (21.0 (IQR = 7.8); *p* < 0.001). Our results showed no significant correlation between Ct values and vaccination status during the Delta and Omicron pandemics. Retrospective analysis revealed that patients receiving dexamethasone treatment had Ct levels significantly below 20 during the Delta wave (*p* < 0.001). This finding indicates that the Ct levels were associated with disease severity in the Delta but not in the Omicron wave. According to age groups, there was no differences between common symptoms in all age groups in either the Delta or Omicron pandemics, whereas the Ct values in all age groups in the Delta pandemic (Ct 20.1 to 21.8) were higher than those in the Omicron pandemic (Ct 18.8 to 20.6) (Appendix A).

### 3.2. Rehospitalized COVID-19 Patients

Eighty-nine (3.3%) and forty-three (1.7%) patients in the HI system were eventually rehospitalized during the Delta and Omicron waves, respectively (Table 2). The mean age of patients with Delta was older than that of patients with Omicron (55 years, IQR = 24 vs. 33 years, IQR = 14; *p* < 0.001). Compared with the Omicron patients, those with Delta had marked lymphocytopenia (0.4-fold) and neutrocytosis (1.8-fold). They also had higher levels of serum C-reactive protein (CRP) (21.2-fold), aspartate aminotransferase (1.6-fold), alanine aminotransferase (2.3-fold), and D-dimer (2-fold) (*p* < 0.05). Vaccination with at least two doses was associated with reduced readmission rates of the Delta patients (odds ratio (OR) = 0.305; 95% confidence interval (CI) 0.189–0.504) and Omicron patients (OR = 0.131; 95% CI 0.052–0.334) than of unvaccinated and partially vaccinated Delta and Omicron patients.

### 3.3. Number of Symptoms and Fatigue Scores during Home Isolation

The risk factors of sex, the severity of illness, and vaccination status were significantly related to increased fatigue, determined by the Chalder fatigue scale (Table 3). Neutralizing antibody titers were independently associated with the number of symptoms (relative risk = 1.22; 95% CI 1.05–1.42; *p* = 0.009). However, the association of neutralizing antibody titers was not statistically significant for fatigue. The severity of the initial illness was associated with persistent fatigue (adjusted relative risk (aRR) = 1.43; *p* = 0.039) and was weakly associated with the number of symptoms (aRR = 1.22; *p* = 0.03). In the stratified analysis of the patients, increased antibody titers remained associated with the number of symptoms (aRR = 1.04; *p* = 0.025) and the fatigue score (aRR = 1.09; *p* = 0.015). Patients who were vaccinated prior to COVID-19 infection reported a significantly lower number of symptoms (*p* < 0.001) and lower fatigue scores (*p* = 0.01) than unvaccinated patients. The Cox proportional hazard model revealed that fever (Exp(B), 0.75; *p* < 0.001), cough (Exp(B), 0.84; *p* < 0.001) and loss of smell (Exp(B), 0.81; *p* < 0.001) were independent risk factors of prolonged NCT of SARS-CoV-2 RNA in patients with COVID-19 (Appendix A).

### 3.4. Clinical Manifestations and Viral Burden

The Ct values decreased markedly in unvaccinated Omicron-dominant individuals (19 (IQR = 17–22)) compared with Delta-dominant individuals (21 (IQR = 18–26); age/sex-adjusted; *p* < 0.001; Figure 2). No difference was observed in individuals vaccinated with either ChAdOx1 or CoronaVac (age/sex-adjusted, *p* = 0.175), indicating that vaccination was still valuable in reducing viral load. During the Omicron wave, new PCR-positive cases were likely to be in the low Ct subpopulation regardless of the number of vaccine doses, the vaccine type, or the time since the last vaccination. However, Ct levels tended to vary during the Delta wave (Figure 3). During the Delta but not the Omicron pandemics, patients who had at least two-dose vaccination prior to COVID-19 infection reported a significantly lower number and probability of any symptoms (OR = 0.25; 95% CI 0.12–0.52; *p* < 0.001) and common COVID-19 symptoms (cough, fever, and anosmia/ageusia (OR = 0.28; 95% CI 0.13–0.58; *p* < 0.001)) than unvaccinated individuals (Figure 4A and Appendix A). No correlation between Ct values and the probability of reporting any symptoms was noted in the Omicron pandemic (Figure 4B–E and Appendix A).

### 3.5. Immune Responses against SARS-CoV-2 Variants

Higher antibody titers were observed in both unvaccinated COVID-19 and vaccinated breakthrough COVID-19 patients. The titers reached their peak around 2 to 3 months post-COVID-19 (PC) and decreased gradually over the following 3 months. The RBD-IgG geometric mean titers (GMT) at baseline (1–2 months PC) were higher in the ChAdOx1 groups (one, two, and three doses) than in the CoronaVac groups. However, no differences in titers were observed between the two groups 3 months PC (ChAdOx1: one dose (822 binding antibody units (BAU)/mL), two doses (945 BAU/mL), and three doses (886 BAU/mL) vs. (CoronaVac: one dose (1174 BAU/mL) and two doses (974 BAU/mL)). In contrast, infected individuals with prior vaccination had higher antibody titers at all time points compared with previously unvaccinated participants with COVID-19 infection (*p* < 0.05; Figure 5A and Appendix A, Appendix A).

The anti-RBD IgG levels and sVNT against the Delta variant were markedly correlated (r = 0.486 to r = 0.599), particularly in the unvaccinated group and vaccinated group (Figure 6 and Appendix A). In both males and females, the GMT of anti-RBD IgG was significantly higher in unvaccinated cases after 2 months (Figure 7A). Older individuals had significantly higher GMT of anti-RBD IgG than the younger individuals in both the unvaccinated and ChAdOx1 groups. Although there was no significant difference in anti-RBD IgG between age groups in the CoronaVac groups, anti-RBD IgG tended to be higher in the older than in the younger individuals (Figure 8A).

Most patients had highly positive sVNT against Wuhan and Delta. Higher sVNT was observed chiefly in breakthrough COVID-19 patients vaccinated with either CoronaVac-prime or ChAdOx1-prime, regardless of the numbers of dose. The titers peaked around 2 to 3 months PC and decreased by approximately 10% to 20% after 3 months, compared with 2 to 3 months PC (Figure 5B and Appendix A). The titers were significantly higher against the Wuhan strain than the Delta variant (*p* < 0.001). Using sVNT, the proportion of individuals with a value for a neutralizing test against Delta above the sVNT cutoff of 30 was approximately 66% to 88% of unvaccinated participants (pink dots). This range contrasted with 83% to 89% for one or two doses of ChAdOx1 (pale blue dots), 100% for three doses of ChAdOx1, and 67% to 95% for one or two doses of CoronaVac.

In fully vaccinated individuals at 2 to 3 months PC, the mean sVNT to the Delta relative to the Wuhan variant was reduced 0.7-fold (from 96.4 to 72.3, ChAdOx1 group). As compared with unvaccinated COVID-19 patients, the mean sVNT for the Delta variant of ChAdOx1-boosted individuals at 2 to 3 months PC were increased 2.8-fold (from 34.7 to 98) and 2.1-fold (from 34.7 to 72.3, compared with two-dose ChAdOx1 group, Figure 5B). In both males and females, the sVNT was significantly higher in unvaccinated cases after 2 months (Figure 7B). The proportion of plasma samples exhibiting such a neutralizing activity against Delta tended to be nearly 1- to 2-fold higher among older than younger individuals (Figure 8B–D and Appendix A).

## 4. Discussion

We confirmed the following central findings. First, illness is mild in most patients, and medical intervention is not needed, particularly in fully vaccinated individuals. These findings confirm that early access to treatment and prompt responses via telehealth visits and antiviral medications provide statistically favorable efficacy in sustaining COVID-19 and improving outcomes in an appropriate outpatient setting [28]. Recent studies demonstrated that patients receiving favipiravir had higher viral clearance rates than patients given standard symptomatic treatment; this higher rate prevented hospitalization [28]. A systematic review and meta-analysis of clinical trials summarized that favipiravir exerted low efficacy in mortality reduction for patients with mild-to-moderate COVID-19. However, the authors also pointed out that that finding might have resulted from delayed treatment in many trials [29].

Second, understanding the relationship between symptoms, viral load, and predictive immunity is crucial to planning for booster vaccination programs. Our results contrast with a recent study [30] reporting lower Ct values for patients infected during the relatively mild Omicron wave than patients infected during the Delta wave. Data showed that primary immunization with two doses of ChAdOx1 or CoronaVac vaccine provided limited protection against symptomatic disease caused by the Delta and the Omicron variants, and vaccine effectiveness waned quickly. Higher neutralizing activity was observed after a booster dose [31]. However, we are the first to show that breakthrough COVID-19 infection with prior vaccination was associated with a significantly lower number of symptoms and fatigue even in the mild-to-moderate COVID-19 disease. On the other hand, the lower the number of symptoms was, the lower the antibody titers were. The difference in risk between the Delta and Omicron waves varied considerably with age but not gender. The risk of hospitalization differed the most for those aged 60 years (50% lower for Omicron compared with Delta) [32]. In contrast, for children under the age of 12 years, there was no significant difference in the risk of hospitalization between Omicron and Delta. The risk of death remained minimal in children. Our unpublished data showed that infected children during the Delta and Omicron pandemics were hospitalized due to insufficient oral intake, anorexia, gastrointestinal tract symptoms, and hypovolemia rather than increased COVID-19 severity.

In addition, in the Omicron pandemic, children made up a larger proportion of patients than in the previous infection waves. One potential explanation for this was that the Omicron variant’s extremely high transmissibility, when coupled with a lack of built-up immunity due to vaccination for 5–11-year-olds in Thailand not being authorized in early 2022 or past infection in young children, left children more vulnerable to Omicron, compared with adults who had access to vaccines for months. Another reason might be that other restrictions and isolation policies were eased at the same time, and parents had the ability to return to the workplace, where transmission also occurred, and immunity waned.

In the mild-to-moderate COVID-19 cases during the Delta and Omicron pandemics, IgG and sVNT were higher in patients with more severe common COVID-19 symptoms. The titers were also associated with high viral loads and older individuals (who were generally vaccinated and had more severe symptoms than asymptomatic individuals) [7]. Higher igG and sVNT was shown in breakthrough COVID-19 patients vaccinated with either CoronaVac-prime or ChAdOx1-primary doses. The titers peaked around 2 to 3 months PC and remained stable for at least 3 months. However, receiving booster vaccines ensured better predictive immunity against COVID-19. In the Omicron pandemic, viral load was not correlated with symptoms. This finding was likely due to Omicron’s milder conditions, an improved vaccination campaign, and quick access to medication treatment. Our results are consistent with the findings of Servellita et al. [7], who examined neutralizing responses in Delta and Omicron breakthrough infections. Substantial increases in antibody titers to Wuhan and Delta were demonstrated, especially after vaccination boosting. In symptomatic or mild Delta and Omicron breakthrough infections, the extent of conferred cross-neutralizing immunity against Omicron and Delta was limited. However, Wratil et al. [33] found that sera from patients with Omicron breakthrough infections significantly enhanced Omicron viral neutralization (17.4-fold).

It is well documented that COVID-19 primarily manifests as a respiratory tract infection, and emerging data indicate it involves multiple systems. Several hematological laboratory investigations have shown that lymphocytes, neutrophils, CRP, elevated D-dimer, and hemostasis are altered significantly in COVID-19 patients [12]. This finding is a potential indicator for both disease progression and the effectiveness of therapy [12]. Evidence indicates that mild COVID-19 may be associated with a potent initial innate antiviral response induction and viral neutralization. These might evade host innate immune activation and, in turn, increase proinflammatory response and immune cell infiltration [34,35]. Even though we did not have a complete set of these parameters for every subject due to the nature of retrospective data from mild-to-moderate COVID-19, we had some patients with worsening conditions who were eventually hospitalized and whose blood was examined. Our results indicated that, during the Delta but not the Omicron wave, these patients had increased neutrophils and lymphocytopenia and activation of the coagulation cascade. However, there were reports that Omicron patients had abnormal levels of neutrophils, lymphocytes, and monocytes and demonstrated signs of coagulopathies [35,36,37]. The adaptive immune response was a key element of the clinical outcome after SARS-CoV-2 infection and supported vaccine efficacy. T-cell responses activated early and correlated with protection but were relatively weakened in severe COVID-19 and were associated with intense activation and lymphopenia [38]. Inflammatory cytokines such as IL-6, IL-8, IL-1β, TNF-α, IFNγ-induced protein10 (IP-10), granulocyte-macrophage colony-stimulating factor (GM-CSF), and chemokines (CC motif) ligand 2 (CCL2), CCL-5, and CCL3 were generally produced by macrophages, mast cells, endothelial and epithelial cells during the innate immune response. Many studies have shown that elevated IL-6 significantly affected the onset of cytokine storm [39,40]. IL-6 played a pleiotropic role in the immune system and was crucial for the formation of TH17 and follicular helper T cells. However, IL-6 could block cytotoxic CD8 + T cells by inhibiting IFN-γ secretion. In addition, IL-6 could impair the cell-induced antiviral response in the cytokine storm. Our unpublished data showed some increased IL-6 levels in some COVID-19 patients. However, we need to clarify the factors predisposing cytokine storms and other inflammatory cytokines. Some studies investigated the T-cell immunity induced after SARS-CoV-2 infection in mild symptomatic cases, showing S-SARS-CoV-2-specific IFN-γ T-cell response was developed [41]. CD4^+^ T-cell responses against SARS-CoV-2 were more prevalent than CD8+ T-cell responses in adults with mild-to-moderate COVID-19 infection [42]. Still, more in-depth research on the underlying etiology is necessary. Gao Y et al. [43] demonstrated that SARS-CoV-2 spike-specific CD4+ and CD8+ T cells elicited by BNT162B2 vaccination or previous infection remain largely intact against the Omicron variant. Together with intrinsic viral factors, these immune reactivities, in part, explain why severe disease appears to be minimal after breakthrough infection with this particular variant.

A recent genome-wide association analysis (GWAS) [44] showed associations of loci on chromosomes 5q32 and 9q21.13 with COVID-19 susceptibility and two suggestive loci on the severity of chromosomes 12q22 and 3p24.3. Interestingly, the association signal on chromosome 5q32 coincided with IL17B encoding a T-cell-derived cytokine known as interleukin-17B (IL-17B). IL-17B was reported to play a role as a proinflammatory inducer in inflammatory disease, stimulating the release of tumor necrosis factor-α (TNF-α) and interleukin-1β (IL-1β) from a monocytic cell line, resulting in neutrophil infiltration [45,46]. This supports our finding of hyperneutrophilia seen in our COVID-19 cohort.

These data combined with ours suggest that the higher infectivity of Omicron may be related to (1) a decreased viral load, (2) probably lower past protective immunity against Omicron (either from vaccines or natural infection with Delta), (3) an asymptomatic stage of infected individuals with respiratory symptoms, and (4) age [20]. The substantial variations in patients’ symptoms and immunogenicity underscore the heterogenicity of protective immunity against future infections. However, an individual previously infected with SARS-CoV-2 is advised to receive a full vaccination course or at least one additional dose of a vaccine after the infection to protect against reinfection from circulating variants [47]. High vaccination rates also help to reduce the transmission of COVID-19. Unfortunately, vaccination rates are still low in some rural areas, important risk groups, and low-income countries [48].

There are several limitations to our study. We only had blood test results from a small subset of hospitalized patients and assumed these findings might be similar to all milder infection cases without hospitalization. No patients with Omicron were treated with Dexamethasone (Favi/Dexa) in the HI system due to the reduced severity of the Omicron infection and increased hospitalization availability for worse cases. However, to compare both waves, we excluded patients in the Favi/dexa group in the Delta wave from the analyses, and the *p*-value was not affected in all parameters. Furthermore, no serology data from patients during the Omicron wave or long-term follow-up data were available for our analysis. Consequently, we could not determine the antibody levels against the Omicron variants, the vaccine efficacy after COVID-19 infection, or the vaccine impact on long COVID-19.

Our future COVID-19 research aims are (1) to gain further insight into the long-term monitoring of neutralizing antibodies and (2) to establish whether breakthrough Omicron infections provide protective immunity against reinfection by SARS-CoV-2 Omicron sub-lineages BA.4 and BA.5.

## 5. Conclusions

Omicron’s mild severity means that a full vaccination course is effective against severe outcomes. Consequently, in countries where vaccine supplies are limited, a full vaccination course with prime or mixed vaccines, and a booster shot for individuals at risk, might be enough to induce high levels of short-term immunity and prevent hospitalization and death. These outcomes should be achievable regardless of a higher viral burden or the symptoms, especially during the Omicron wave in the absence of novel variants.

## Figures and Tables

**Figure 1 vaccines-10-01131-f001:**
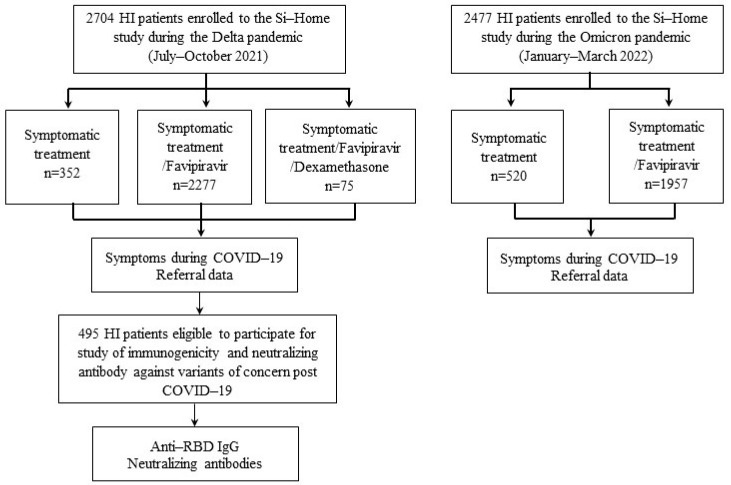
Study source recruitment and enrollment.

**Figure 2 vaccines-10-01131-f002:**
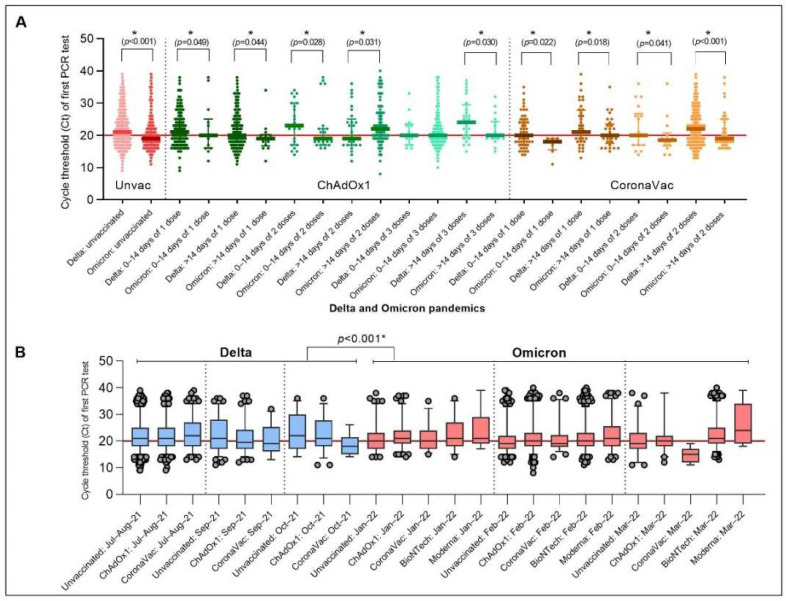
Ct value trajectories of confirmed COVID-19 infection and symptoms during the Delta and Omicron pandemics in vaccinated and unvaccinated individuals. (**A**) Ct values by vaccination/reinfection status and (**B**) Ct values by waves and vaccination type. * *p* < 0.05.

**Figure 3 vaccines-10-01131-f003:**
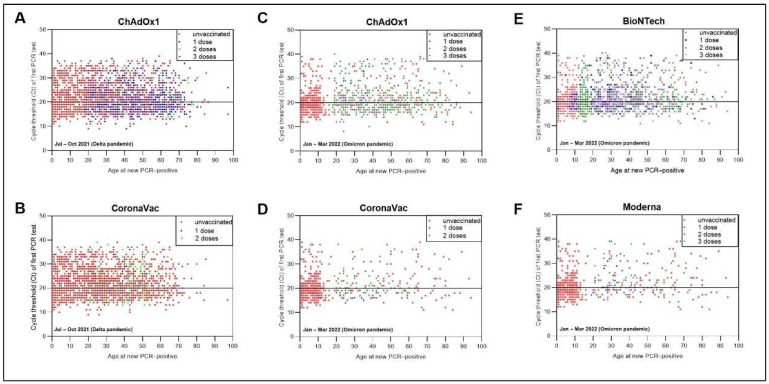
Ct value trajectories of confirmed COVID-19 infection and symptoms during the Delta (**A**,**B**) and Omicron (**C**–**F**) pandemics in vaccinated and unvaccinated individuals. Ct values in PCR-positives after receiving ChAdOx1 or CoronaVac or BioNTech or Moderna vaccines or unvaccinated, regardless of vaccine doses by time since July 2021. Red dots are represented in all figures as reference values.

**Figure 4 vaccines-10-01131-f004:**
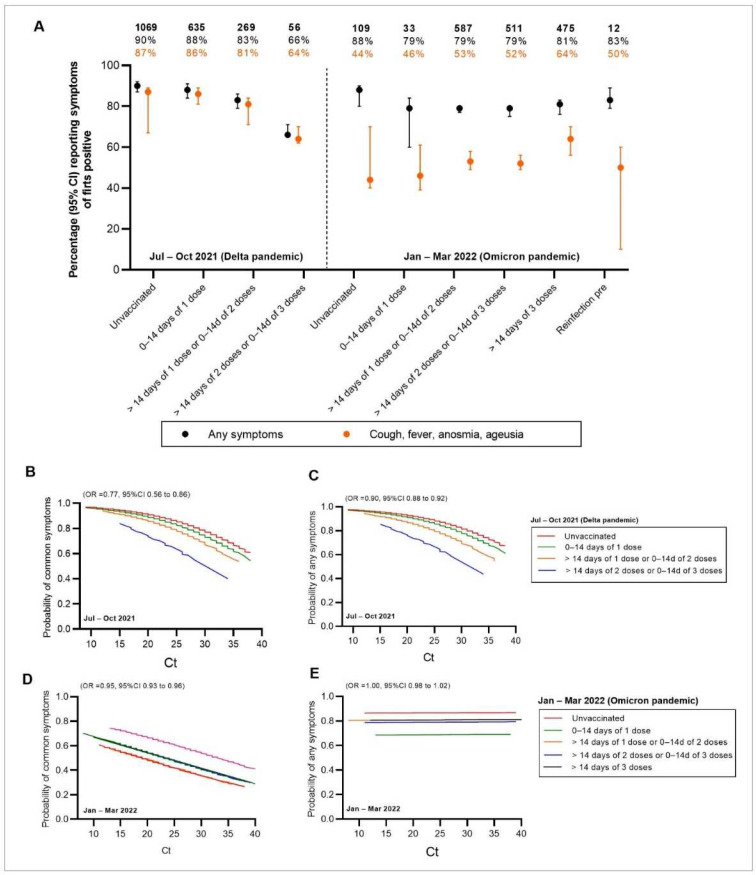
Ct value trajectories of confirmed COVID-19 infection and symptoms during the Delta and Omicron pandemics in vaccinated and unvaccinated individuals. (**A**) Self-report symptoms in PCR-positives by numbers of vaccination/reinfection status. Probability of reporting common (**B**,**D**) fever, cough, anosmia, or ageusia or (**C**,**E**) any symptoms by Ct values and vaccination status in PCR-positives.

**Figure 5 vaccines-10-01131-f005:**
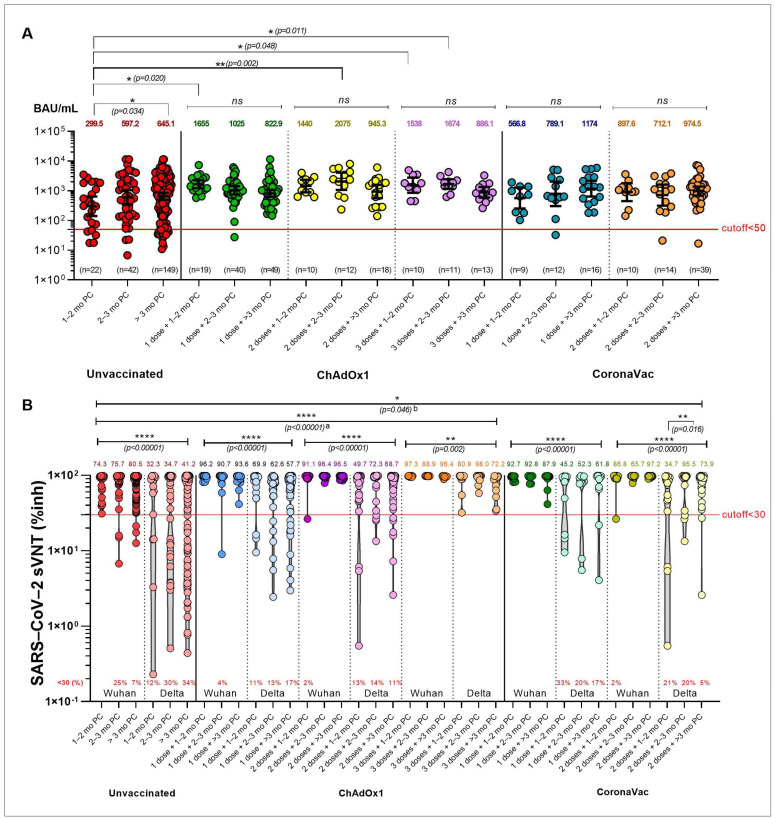
Immune responses after breakthrough COVID-19 infection with prior CoronaVac or ChAdOx1 vaccination during the Delta pandemic. (**A**) The scatter plot of geometric mean titers (GMTs) of SARS-CoV-2 anti-spike protein receptor-binding domain antibodies’ (Anti-RBD IgG) concentrations in serum samples obtained from subjects after COVID-19 infection and with prior various vaccination status (CoronaVac vs. ChAdOx1). Sera at different time points from patients recovered from COVID-19 are shown as reference level (red). (**B**) Scatter plots demonstrate an inhibition rate of Wuhan and Delta RBD-blocking antibodies measured using a surrogate viral neutralization test (sVNT) by vaccination/reinfection status; the lower dot line represents the cut-off level for seropositivity. All sera were from the patients during the Delta pandemic. * *p* < 0.05; ** *p* < 0.01; **** *p* < 0.0001.

**Figure 6 vaccines-10-01131-f006:**
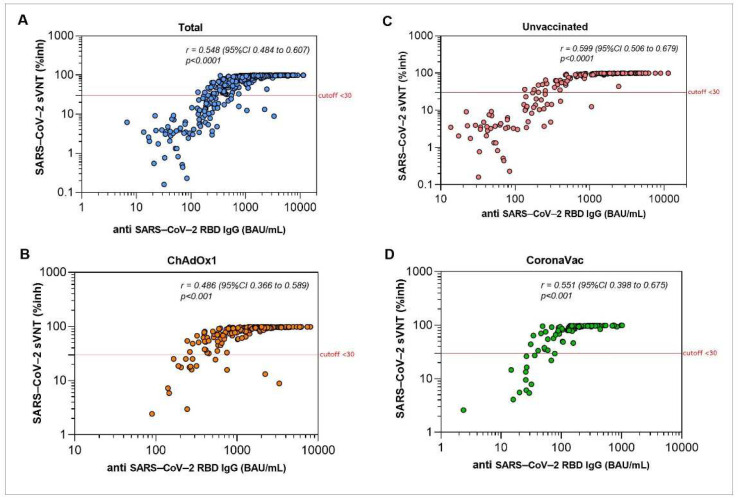
Immune responses after breakthrough COVID-19 infection with prior CoronaVac or ChAdOx1 vaccination during the Delta pandemic. Dot plots show the correlation between the level of anti-SARS-CoV-2 RBD IgG and surrogate viral neutralization test (sVNT) for the SARS-CoV-2 delta variant in plasma of study participants (total, (**A**) who were unvaccinated (**C**), or completed two doses of ChAdOxX1 (**B**), CoronaVac (**D**) and had breakthrough infection.

**Figure 7 vaccines-10-01131-f007:**
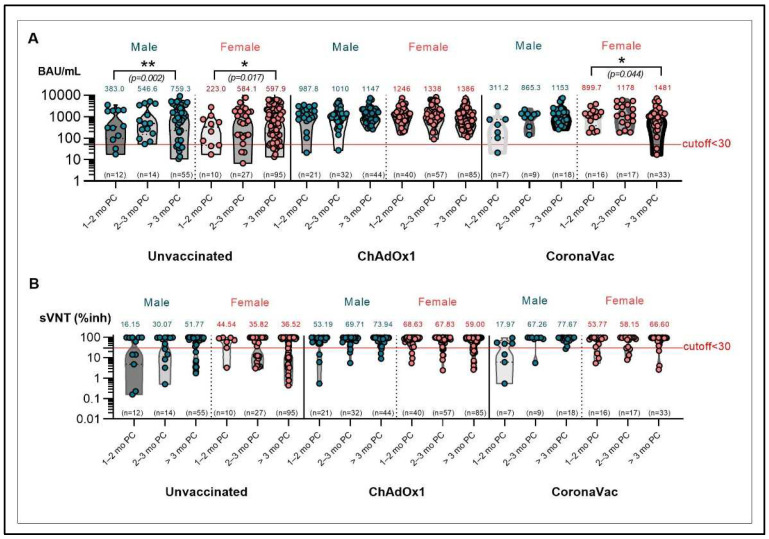
Immune responses after breakthrough COVID-19 infection with prior CoronaVac or ChAdOx1 vaccination during the Delta pandemic. Anti-RBD IgG concentrations (**A**), sVNT (**B**) by sex/vaccination/reinfection status. * *p* < 0.05; ** *p* < 0.01.

**Figure 8 vaccines-10-01131-f008:**
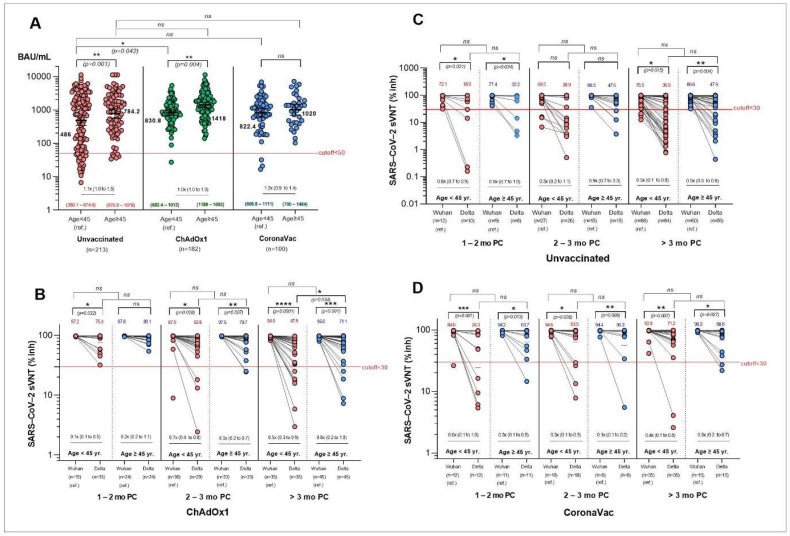
Immune responses after breakthrough COVID-19 infection with prior CoronaVac or ChAdOx1 vaccination during the Delta pandemic. Antibodies (Anti-RBD IgG) (**A**) by age/vaccination, and sVNT (**B**–**D**) by age groups/vaccination/infection status. * *p* < 0.05; ** *p* < 0.01; *** *p* < 0.001; **** *p* < 0.0001.

**Table 1 vaccines-10-01131-t001:** Characteristics and laboratory findings of all confirmed COVID-19 patients, compared between those with symptomatic treatment (S), symptomatic treatment plus 5–14 days’ standard favipiravir treatment (Favi) and symptomatic treatment plus 5–14 days’ standard favipiravir treatment plus dexamethasone treatment (Favi/Dexa) *.

Characteristics	July–October 2021 (Delta)									January–March 2022 (Omicron)							*p* *^,§^
	All Patients (*n* = 2704)		S (*n* = 352)		Favi (*n* = 2277)		Favi/Dexa (*n* = 75)		*p* *^,†^	All Patients (*n* = 2477)		S (*n* = 520)		Favi (*n* = 1957)		*p* *^,‡^	
	*n*	%	*n*	%	*n*	%	*n*	%		*n*	%	*n*	%	*n*	%		
Female sex	1451	53.7	176	50.0	1235	54.2	40	53.3	0.332	1446	58.4	340	65.4	1106	56.5	0.020	0.001
Age, year, mean (SD)	33.8	(11.6)	15.7	(9.4)	36.0	(8.7)	52.5	(6.4)	<0.001	31.3	(12.3)	36.0	(15.4)	30.1	(11.2)	<0.001	<0.001
<25	955	35.3	275	78.1	677	29.7	3	4.0	<0.001	1015	41.0	115	22.1	900	46.0	<0.001	<0.001
25–60	1470	54.4	75	21.3	1350	59.3	45	60.0		1220	49.3	360	69.2	860	43.9		
>60	279	10.3	2	0.6	250	11.0	27	36.0		242	9.8	45	8.7	197	10.1		
Body weight, kg, mean (SD)	57.5	(22.0)	38.3	(21.4)	59.7	(20.9)	69.7	(13.2)	0.043	55.8	(22.4)	61.6	(16.7)	54.2	(23.4)	0.034	0.007
Presence of comorbidities																	
Diabetes mellitus	162	6.0	1	0.3	151	6.6	10	13.3	<0.001	150	6.1	34	6.5	116	5.9	0.604	0.922
Hypertension	313	11.6	7	2.0	283	12.4	23	30.7	<0.001	513	20.7	116	22.3	397	20.3	0.312	<0.001
Dyslipidemia	123	4.5	2	0.6	113	5.0	8	10.7	0.017	108	4.4	16	3.1	92	4.7	0.107	0.742
Obesity	23	0.9	0	0.0	23	1.0	0	0.0	0.114	41	1.7	7	1.3	34	1.7	0.534	0.009
Malignancy	22	0.8	0	0.0	21	0.9	1	1.3	0.176	25	1.0	2	0.4	23	1.2	0.109	0.458
Neurologic disease	7	0.3	0	0.0	7	0.3	0	0.0	0.288	271	10.9	78	15.0	193	9.9	0.001	<0.001
Heart disease	33	1.2	1	0.3	31	1.4	1	1.3	0.230	38	1.5	2	0.4	36	1.8	0.016	0.332
Lung disease	51	1.9	1	0.3	30	1.3	20	26.7	<0.001	na		na		na			
Kidney disease	14	0.5	1	0.3	13	0.6	0	0.0	0.641	10	0.4	1	0.2	9	0.5	0.392	0.546
Others	336	12.4	33	9.4	292	12.8	11	14.7	0.158	753	30.4	162	31.2	591	30.2	0.674	<0.001
Presenting symptoms																	
Asymptomatic infection	390	14.4	103	29.3	286	12.6	1	1.3	<0.001	969	39.1	224	43.1	745	38.1	0.038	<0.001
Fever/history of fever	1250	46.2	133	37.8	1074	47.2	43	57.3	0.001	267	10.8	23	4.4	244	12.5	<0.001	<0.001
BT ^¶^ (°C), median (IQR) ^‖^	36.6	(0.7)	36.3	(0.6)	36.6	(0.7)	37.0	(0.6)	0.015	36.8	(0.5)	36.7	(0.4)	36.9	(0.5)	0.149	<0.001
<37.5	2474	95.2	318	97.0	2094	95.3	62	83.8	0.002	2107	88.8	489	95.5	1618	86.9	0.015	<0.001
37.5–38.0	120	4.6	10	3.0	99	4.5	11	14.9		244	10.3	21	4.1	223	12.0		
>38.0	6	0.2	0	0.0	5	0.2	1	1.4		23	1.0	2	0.4	21	1.1		
Cough	1642	60.7	152	43.2	1425	62.6	65	86.7	<0.001	1181	47.7	242	46.5	939	48.0	0.558	<0.001
Sore throat	1038	38.4	81	23.0	918	40.3	39	52.0	0.010	1181	47.7	242	46.5	939	48.0	0.558	<0.001
Rhinorrhea	419	15.5	47	13.4	358	15.7	14	18.7	0.383	626	25.3	143	27.5	483	24.7	0.189	<0.001
Productive sputum	537	19.9	46	13.1	465	20.4	26	34.7	0.029	na		na		na			
Loss of taste	312	11.5	23	6.5	274	12.0	15	20.0	0.010	43	1.7	12	2.3	31	1.6	0.261	<0.001
Loss of smell	821	30.4	66	18.8	725	31.8	30	40.0	0.017	43	1.7	12	2.3	31	1.6	0.261	<0.001
Dyspnea	305	11.3	5	1.4	271	11.9	29	38.7	0.012	24	1.0	2	0.4	22	1.1	0.126	<0.001
Myalgia	282	10.4	11	3.1	249	10.9	22	29.3	0.005	206	8.3	52	10.0	154	7.9	0.118	0.009
Diarrhea	126	4.7	11	3.1	108	4.7	7	9.3	<0.001	63	2.5	10	1.9	53	2.7	0.312	<0.001
Nausea/vomiting	59	2.2	2	0.6	49	2.2	8	10.7	0.038	37	1.5	5	1.0	32	1.6	0.260	0.067
Others	1549	57.3	168	47.7	1328	58.3	53	70.7	<0.001	191	7.7	36	6.9	155	7.9	0.449	<0.001
Clinical features at the time of admission																	
Time from symptom onset to PCR diagnosis, median (IQR), days	1.9	(1.6)	2.1	(1.9)	1.9	(1.6)	1.9	(1.5)	0.394	2.0	(1.1)	2.1	(1.0)	2.0	(1.1)	0.167	0.049
Time from symptom onset to admission, median (IQR), days	5.1	(2.4)	5.6	(2.6)	5.0	(2.3)	5.2	(2.0)	0.337	2.8	(1.6)	3.2	(2.0)	2.6	(1.4)	0.032	0.021
Cycle threshold **																	
Cycle threshold, median (IQR)	21.0	(7.8)	24.1	(9.4)	20.7	(7.4)	18.5	(4.3)	0.001	19.0	(5.7)	21.4	(6.9)	19.7	(5.3)	0.001	<0.001
<20	1118	43.1	88	25.9	979	45.0	51	68.9	<0.001	1162	49.4	201	38.7	961	52.4	<0.001	<0.001
20–30	1218	47.0	189	55.6	1008	46.3	21	28.4		974	41.4	240	46.2	734	40.0		
>30	255	9.8	63	18.5	190	8.7	2	2.7		216	9.2	78	15.0	138	7.5		
Envelope (E), median (IQR)	17.5	(8.4)	20.9	(9.9)	17.3	(8.0)	14.3	(4.7)	0.012	18.0	(5.4)	17.8	(5.1)	19.0	(6.7)	0.121	<0.001
RNA-dependent RNA polymerase (RdRp), median (IQR)	22.3	(8.2)	25.9	(9.6)	22.0	(7.7)	19.7	(4.7)	<0.001	19.3	(5.4)	20.3	(6.9)	19.1	(5.0)	0.220	<0.001
Referred back to the hospital, yes	89	3.3	3	0.9	61	2.7	25	33.3	<0.001	43	1.7	5	1.0	38	1.9	0.128	<0.001
Dead, yes	5	0.2	0	0.0	0	0.0	5	6.7	<0.001	2	0.1	0	0.0	2	0.1	0.466	0.308

* Continuous data of characteristics and laboratory findings of all confirmed COVID-19 patients presented as mean (SD), median (IQR), and range at *p* < 0.05 indicates statistical significance. ^†^ The statistical significance was assessed using the Fisher’s exact test and Kruskal–Wallis test; statistical difference within the Delta group was at *p* < 0.05. ^‡^ The statistical significance was assessed using the Fisher’s exact test and Mann–Whitney test statistical difference within the Omicron group at *p* < 0.05. ^§^ The statistical significance was assessed using the Fisher’s exact test and Mann–Whitney test; statistical difference between Delta and Omicron groups was at *p* < 0.05. ^¶^ Body temperature (BT) is a measure of the balance between heat generation and heat loss of the body. ^‖^ Interquartile range (IQR) is a measure of statistical dispersion. ** Cycle threshold (Ct) value from RT-PCR tests represents the cycle number at which the signal breaches the threshold for positivity; a lower Ct value is indicative of a high viral load.

**Table 2 vaccines-10-01131-t002:** Demographic, clinical, and laboratory findings of all of patients referred back for in-patient care and compared between those during the Delta and the Omicron pandemic *.

Characteristics	July–October 2021 (Delta)							January–March 2022 (Omicron)							*p* *^,§^
	All Patients (*n* = 89)		Alive (*n* = 84)		Dead (*n* = 5)		*p* *^,†^	All Patients (*n* = 43)		Alive (*n* = 41)		Dead (*n* = 2)		*p* *^,‡^	
	*n*	%	*n*	%	*n*	%		*n*	%	*n*	%	*n*	%		
Male sex	44	49.4	43	51.2	1	20.0	0.175	20	46.5	18	43.9	2	100.0	0.121	0.753
Age, year, median (IQR)	55.0	(24.0)	55.5	(24.5)	54.0	(5.0)	0.617	33.0	(14.0)	30.5	(12.0)	81.0	(8.5)	0.073	<0.001
<18	5	5.6	5	6.0	0	0.0	0.017	25	58.1	25	61.0	0	0.0	0.022	0.697
18–44	26	29.2	26	31.0	0	0.0		4	9.3	4	9.8	0	0.0		
45–64	37	41.6	32	38.1	5	100.0		3	7.0	3	7.3	0	0.0		
≥65	21	23.6	21	25.0	0	0.0		11	25.6	9	22.0	2	100.0		
Presence of comorbidities															
Diabetes mellitus	23	25.8	21	25.0	2	40.0	0.457	7	16.3	7	17.1	0	0.0	na	<0.001
Hypertension	28	31.5	26	31.0	2	40.0	0.672	10	23.3	10	24.4	0	0.0	na	<0.001
Dyslipidemia	12	13.5	12	14.3	0	0.0	0.364	6	14.0	6	14.6	0	0.0	na	<0.001
Heart disease	na		na		na			3	7.0	3	7.3	0	0.0	na	<0.002
Others	31	34.8	28	33.3	3	60.0	0.224	2	4.7	2	4.9	0	0.0	na	0.258
Presenting symptoms of entering HI ^¶^															
Asymptomatic infection	9	10.1	9	10.7	0	0.0	0.440	32	74.4	30	73.2	2	100.0	0.396	0.020
Fever/history of fever	49	55.1	46	54.8	3	60.0	0.464	27	62.8	25	61.0	2	100.0	0.530	0.003
BT ^‖^ (°C), median (IQR) **	36.8	(0.5)	36.8	(0.5)	36.8	(0.2)	0.156	36.7	(0.9)	36.7	(0.9)	-	-	na	
>38.0	9	10.1	8	9.5	1	20.0	0.429	2	4.7	2	4.9	0	0.0	na	0.660
Cough	61	68.5	58	69.0	3	60.0	0.672	20	46.5	18	43.9	2	100.0	0.258	0.533
URI ^††^	45	50.6	43	51.2	2	40.0	0.489	21	48.8	21	51.2	0	0.0	0.111	0.174
Loss of taste/smell	22	24.7	21	25.0	1	20.0	0.201	1	2.3	1	2.4	0	0.0	0.793	0.008
Dyspnea	53	59.6	48	57.1	5	100.0	0.013	2	4.7	2	4.9	0	0.0	0.706	<0.001
Muscle aches	23	25.8	21	25.0	2	40.0	0.101	8	18.6	8	19.5	0	0.0	0.399	0.925
Diarrhea	13	14.6	11	13.1	2	40.0	0.098	2	4.7	2	4.9	0	0.0	0.706	0.219
Nausea/vomiting	10	11.2	10	11.9	1	20.0	0.413	6	14.0	6	14.6	0	0.0	0.483	0.372
Chest radiograph on referral date															
Pneumonia detected in chest radiograph	51	57.3	48	57.1	3	60.0	0.638	12	27.9	12	29.3	0	0.0	na	0.158
Hematological, median (IQR)															
WBC ^‡‡^ (×10^3^/µL)	7.7	(7.4)	7.7	(7.4)	7.4	(7.2)	0.785	6.3	(4.0)	6.3	(3.7)	8.8	(9.8)	0.061	0.012
Lymphocytes (×10^3^/µL)	0.8	(0.8)	0.9	(0.4)	0.8	(0.8)	0.038	1.8	(2.1)	1.8	(2.0)	2.1	(2.6)	0.914	<0.001
<1 × 10^3^/uL	35	39.3	32	38.1	3	60.0	0.592	12	27.9	11	26.8	1	50.0	0.492	0.001
Neutrophil (×10^3^/µL)	6.1	(7.0)	6.4	(7.4)	6.1	(7.0)	0.584	3.4	(2.5)	3.4	(2.6)	6.1	(7.3)	0.047	<0.001
Creatinine (mg/dL)	0.8	(0.4)	0.8	(0.4)	1.0	(1.6)	0.068	0.8	(0.8)	0.8	(0.9)	0.8	(0.3)	0.047	<0.001
eGFR ^§§^ (mL/min/1.73 m^2^)	84.5	(37.5)	84.5	(38.2)	56.4	(87.1)	0.080	67.0	(64.5)	67.0	(64.5)	-	-	na	0.139
AST ^¶¶^ (U/L)	47.5	(27.5)	47.0	(26.0)	59.0	(146.0)	0.043	30.0	(17.0)	30.0	(17.0)	36.0	-	na	<0.001
>40	35	39.3	32	38.1	3	60.0	0.056	10	23.3	10	24.4	0	0.0	0.552	<0.001
ALT ^¶¶^ (U/L)	41.5	(32.5)	42.0	(32.0)	19.0	(39.0)	0.337	18.0	(8.0)	17.5	(8.0)	22.0	-	na	<0.001
>40	29	32.6	28	33.3	1	20.0	0.511	3	7.0	3	7.3	0	0.0	0.771	<0.001
C-reactive protein (mg/L)	72.1	(65.2)	72.5	(64.0)	64.1	(54.7)	0.154	3.4	(9.2)	3.4	(8.7)	67.6	(29.3)	0.035	0.008
<10	6	9.5	4	6.7	2	66.7	0.005	31	73.8	30	75.0	1	50.0	0.114	<0.001
10–100	40	63.5	40	66.7	0	0.0		8	19.0	8	20.0	0	0.0		
>100	17	27.0	16	26.7	1	33.3		3	7.1	2	5.0	1	50.0		
Procalcitonin (ng/mL)	0.1	(0.3)	0.1	(0.2)	0.5	(1.5)	0.085	0.1	(0.3)	0.1	(0.3)	0.1	(0.2)	0.439	<0.001
>0.05	40	83.3	36	81.8	4	100.0	0.097	9	69.2	7	63.6	2	100.0	0.305	0.257
D-dimer, median (ng/mL)	2671.9	(1008.9)	2552.1	(1067.1)	4308.0	(2872.6)	0.059	1347.0	(4588.5)	1297.0	(2723.0)	7633.0	-	na	0.029
<500	13	29.5	13	31.7	0	0.0	0.215	0	0.0	0	0.0	0	0.0	0.117	0.099
501–3000	25	56.8	23	56.1	2	66.7		5	62.5	5	71.4	0	0.0		
>3000	6	13.6	5	12.2	1	33.3		3	37.5	2	28.6	1	100.0		
Cycle Threshold ^‖‖^ (viral load at the time of entering HI)															
Nucleocapsid (N), median (IQR)	19.1	(5.4)	19.2	(5.2)	16.1	(3.7)	0.846	18.9	(3.9)	18.9	(3.2)	18.2	(5.6)	0.729	0.023
<20	57	64.0	53	63.1	4	80.0	0.714	31	72.1	30	73.2	1	50.0	0.567	0.414
20–30	28	31.5	27	32.1	1	20.0		9	20.9	8	19.5	1	50.0		
>30	4	4.5	4	4.8	0	0.0		3	7.0	3	7.3	0	0.0		
Vaccine status															
Unvaccinated	58	65.2	54	64.3	4	80.0	0.408	29	67.4	27	65.9	2	100.0	0.603	0.134
1 dose	10	11.2	9	10.7	1	20.0		9	20.9	9	22.0	0	0.0		
≥2 doses	21	23.6	21	25.0	0	0.0		5	11.6	5	12.2	0	0.0		
ORcrude (95% CI) ***	0.309 (0.189–0.504)							0.131 (0.052–0.334)							
ORage and sex adjusted	0.142 (0.016–0.265)							0.109 (0.011–0.318)							
ORfully adjusted	0.299 (0.012–7.643)							0.105 (0.005–0.294)							

* Continuous data of demographic, clinical, and laboratory findings of all patients referred back presented as mean (SD), median (IQR), and range at *p* < 0.05 which indicates statistical significance; OR, odds ratio. ^†^ The statistical significance was assessed using the Fisher’s exact test and Kruskal–Wallis test; statistical difference within the Delta group was at *p* < 0.05. ^‡^ The statistical significance was assessed using the Fisher’s exact test and Mann–Whitney test; statistical difference within the Omicron group was at *p* < 0.05. ^§^ The statistical significance was assessed using the Fisher’s exact test and Mann–Whitney test; statistical difference between Delta and Omicron groups was at *p* < 0.05. ^¶^ Home isolation (HI): Once a COVID-19 infection has been diagnosed, medical staff will assess home isolation. The patients should generally be in good health and should not be suffering from any of the following conditions: chronic obstructive pulmonary disease (COPD), chronic kidney disease (CKD), cardiovascular disease, cerebrovascular disease, uncontrollable diabetes, or other conditions that may be considered by doctors to be a risk. Patients must agree to strictly isolate themselves from others. ^‖^ Body temperature (BT) is a measure of the balance between heat generation and heat loss of the body. ** Interquartile range (IQR) is a measure of statistical dispersion. ^††^ Upper respiratory infection (URI) affects the upper part of your respiratory system. ^‡‡^ White blood count (WBC) is part of the immune system, helping to defend the body against infections and disease. ^§§^ Estimated Glomerular Filtration Rate (eGFR) is used to determine if one has kidney disease. ^¶¶^ Aspartate aminotransferase (AST) is an enzyme that is present in various tissues of the body, while alanine aminotransferase (ALT) is found mainly in your liver, used to check for liver conditions, while AST is found in more parts of the body than ALT. For this reason, abnormal levels of ALT tend to be better indicators of liver problems than AST. ^‖‖^ Cycle threshold (Ct) value from RT-PCR tests represents the cycle number at which the signal breaches the threshold for positivity; a lower Ct value is indicative of a high viral load. *** Effect estimates are reported as ORs (95% CIs); unvaccinated and 1-dose (reference) groups vs. >2 doses were compared by using multivariable logistic regression to calculate adjusted ORs (aORs) with 95% CIs.

**Table 3 vaccines-10-01131-t003:** COVID-19 patient factors associated with increasing number of symptoms and higher fatigue score at 14-day follow-up—negative binomial mixed models (July 2021 to March 2022) *.

Characteristics	*n* %		Number of Symptoms ^†^ (0–12)						Fatigue Score ^‡^ (0–27)					
			RR ^§^	95% CI ^¶^	*p* *^,‖^	aRR **	95% CI	*p* *^,‖^	RR	95% CI	*p* *^,††^	aRR	95% CI	*p* *^,††^
Female sex	2194	58.4	1.02	(0.94, 1.10)	0.650	1.03	(0.96, 1.11)	0.361	1.13	(0.99, 1.30)	0.071	1.15	(1.00, 1.31)	0.049 *
Age, year, median (range)	31 (17–47)		1.00	(1.00, 1.00)	0.966	1.00	(1.00, 1.00)	0.146	1.00	(0.99, 1.00)	0.458	1.00	(0.99, 1.00)	0.292
Comorbidity														
Hypertension	769	20.5	1.05	(0.95, 1.16)	0.322	1.04	(0.93, 1.16)	0.488	0.97	(0.81, 1.15)	0.708	-	-	-
Dyslipidemia	231	6.2	1.01	(0.89, 1.15)	0.890	1.07	(0.93, 1.22)	0.353	0.97	(0.77, 1.22)	0.781	1.11	(0.86, 1.42)	0.427
Diabetes mellitus	311	8.3	0.99	(0.87, 1.13)	0.914	0.95	(0.83, 1.09)	0.491	0.83	(0.65, 1.05)	0.115	0.81	(0.63, 1.04)	0.104
Asthma/COPD ^‡‡^	97	4.8	0.99	(0.85, 1.15)	0.882	0.99	(0.86, 1.14)	0.882	0.84	(0.64, 1.10)	0.211	0.85	(0.65, 1.11)	0.232
Chronic heart disease	65	1.7	1.07	(0.83, 1.39)	0.592	1.07	(0.84, 1.37)	0.568	1.22	(0.77, 1.93)	0.39	1.30	(0.83, 2.04)	0.252
Severity of initial illness	102	2.7	1.34	(1.11, 1.61)	0.002 *	1.22	(1.02, 1.45)	0.030 *	1.46	(0.95, 1.75)	0.051	1.43	(1.37, 1.72)	0.039 *
Immunosuppression ^§§^, median (range)	8.89 (7.94–9.53)		1.05	(1.03, 1.08)	<0.001 *	1.04	(1.01, 1.08)	0.025 *	1.04	(1.00, 1.10)	0.074	1.09	(1.02, 1.16)	0.015 *
Neutralizing antibody titers ^¶¶^, median (range)	4.58 (4.55–4.59)		1.22	(1.08, 1.37)	0.001 *	1.22	(1.05, 1.42)	0.009 *	1.18	(0.80, 1.20)	0.876	1.14	(0.66, 1.19)	0.324
Vaccinated, yes	2577	68.6	0.81	(0.76, 0.87)	<0.001 *	0.78	(0.72, 0.84)	<0.001 *	0.85	(0.75, 0.97)	0.014 *	0.84	(0.73, 0.96)	0.011 *

* Analysis of associated factors was conducted using negative binomial mixed models. RR, relative risk; aRR, adjusted relative risk. Statistical significance at the level of *p* < 0.05 is shown in bold text. ^†^ Patients were assessed for 12 symptoms mentioned in Appendix A. ^‡^ Chalder fatigue score is only validated for patients aged >18 years (*n* = 3756); possible fatigue scores range from 0 (no fatigue) to 27 (worst possible fatigue). ^§^ The relative risk (RR) is the risk of an event in an experimental group relative to that in a control group. ^¶^ The 95% confidence interval (CI) is used to estimate the precision of the OR. ^‖^ Factors with significance level *p* < 0.05 in univariable analysis were included in the multivariable analysis of symptoms at 14-day follow-up. ** Adjusted relative risk (aRR) is the difference in increased risk of symptoms and fatigue score. ^††^ Factors with significance level *p* < 0.05 in univariable analysis were included in the multivariable analysis of fatigue score at 14-day follow-up. ^‡‡^ COPD is chronic obstructive pulmonary disease. ^§§^ SARS-CoV-2 spike protein antibody titers, log10 transformed. ^¶¶^ Neutralizing antibody titers, log10 transformed.

## Data Availability

Raw data used in this study, including de-identified patient metadata and test results, are available upon request.

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
