# Peer review of "SARS-CoV-2 Antibody Response against Mild-to-Moderate Breakthrough COVID-19 in Home Isolation Setting in Thailand"

_vaccines, 2022, doi:10.3390/vaccines10071131_

Round 1

Reviewer 1 Report

The topic of the present study is interesting, given the great concern about the immunity response to different SARS-CoV-2 variants. In spite of this, revisions are required as follows.

Introduction

It is not exaustive. Informations on both vaccines used in this study (CoronaVac and ChAdOx1) should be added: type (DNA, RNA), mechanism of action, references. Favipiravir information is also required: action, reference, antiviral use (against RNA virus infections: anti-influenza drug, anti Covid-19). To introduce the work, a brief presentation on the difference in patients’ symptoms and the vaccine’s effectiveness against the new variants should be made.

Materials and Methods

Figure S2 shoud be moved here in the main text for a better comprehension of the procedure.

A focal point is the diagnostic assay. Please explain better the approach used, not only in S1 but also in the main text. Describe which RT-PCR probes were used to detect Delta and Omicron variants. This is also very important with respect to serological analysis (performed on Whuan and Delta but not on Omicron).

Write in full the meaning of the abbreviations sVNT, Ct.

Results

Paragraph 3.1. The number of patients in each age class should be considered, as well as the incidence. In addition, these data may be due to individual susceptibility to virus variants, different vaccine type, reduced isolation measures, etc., and need to be discussed.

No patients with Omicron were treated with dexamethasone (Favi/Dexa): how to compare the data with Delta patients?

Tables 1, 2, and 3: uniform text and spacing characters in notes.

Table 3. Write in full the meaning of the abbreviation RR.

The figures are too small to display, especially fig. 2 C-L. I suggest reducing the number of figures, choosing the most significant ones for a better understanding of the results.

Lines 250-252. I suggest reporting the ages in the text. Age are not shown in figure S4, so why is it cited here?

Discussion

In addition to some points as previously indicated, a more in depth discussion is needed. There has been a lot of data recorded on symptoms in respect to vaccine status as well as to age and gender, so they deserve a discussion of differences observed in respect to all of these parameters. This will help to better support conclusions.

Supplementary

Page 4. X” vaccination”, capital letter.

Page 9. I did not find the note “7” in the text.

Author Response

Response to Reviewer 1 Comments

The topic of the present study is interesting, given the great concern about the immunity response to different SARS-CoV-2 variants. In spite of this, revisions are required as follows.

Point 1: Introduction

It is not exaustive. Informations on both vaccines used in this study (CoronaVac and ChAdOx1) should be added: type (DNA, RNA), mechanism of action, references. Favipiravir information is also required: action, reference, antiviral use (against RNA virus infections: anti-influenza drug, anti Covid-19). To introduce the work, a brief presentation on the difference in patients’ symptoms and the vaccine’s effectiveness against the new variants should be made.

Response 1: We have addressed the comments carefully.

We have added “CoronaVac (a whole-cell inactivated vaccines, Sinovac, Life Science) and ChAdOx1 (a modified chimpanzee DNA adenovirus-vectored vaccine, AstraZeneca/Oxford) were more widely used than other vaccines by Thais [3]. CoronaVac vaccine is based on a form of severe acute respiratory syndrome coronavirus 2 (SARS-CoV-2) that has been weakened and safely generate an immune response [4]. AstraZeneca vaccine is based on the virus’s genetic instructions containing the SARS-CoV-2 structural surface glycoprotein antigen (spike protein; nCoV-19) gene for building the spike protein. The spike protein fragments can then be recognized by the immune system [4,5]. Both vaccines were efficacious against symptomatic COVID-19 caused by the Wuhan strain but…”. (Page 2, Line 58-66)

“Vaccine effectiveness is low and wanes faster against infection and mild-to-moderate symptomatic disease but is high against severe disease caused by the Omicron variant. Evidence indicated that vaccine effectiveness against severe disease outcome after receipt of a primary series with either CoronaVac or AstraZeneca or a booster dose increased to > 70% for all vaccines within the first 3 months after a final dose [8].” (Page 2, Line 68-73)

“Since 2020, the Thai National Treatment Guidelines for COVID-19 Ministry of Public Health [9] recommended that favipiravir, a broad-spectrum nucleotide analog targeting the viral RNA-dependent RNA polymerase [10], be the treatment option for patients at increased risk of severe disease and mild severity of pneumonia. It has widely been repurposed to treat mild-to-moderate cases of COVID-19, including Delta and Omicron. Based on our experiences and in earlier studies, it showed promising re-sults in patients with mild-to-moderate COVID-19 with well-tolerated side effects [11,12]. Remdesivir is recommended only for severe disease due to limited access. In addition, monipiravir, the oral prodrug of beta-D-N4-hydroxycytidine (NHC) and an-ti-SARS-CoV-2 monoclonal antibodies were not available during the study period.” (Page 2, Line 78-87)

Point 2: Materials and Methods

- Figure S2 should be moved here in the main text for a better comprehension of the procedure.

- A focal point is the diagnostic assay. Please explain better the approach used, not only in S1 but also in the main text. Describe which RT-PCR probes were used to detect Delta and Omicron variants. This is also very important with respect to serological analysis (performed on Wuhan and Delta but not on Omicron).

- Write in full the meaning of the abbreviations sVNT, Ct.

Response 2: We have addressed the reviewer’s comments.

We have moved Figure S2 to be Figure 1 in the main text (Page 3).

We have added details about outcome measures and serological assays in the highlighted sentences in Outcome Measures Section, Serological Assays Section and Statistical Analysis Section.

2.3. Outcome Measures

The rationale for the mild-to-moderate COVID-19 treatment is described in Method S1. In brief, the primary treatment strategy in Thailand included early Favipiravir treatment and recommended outpatient antiviral therapies. (Page 4, Line 133-135)

The “date of disease onset” was defined as the day when new-onset self-reported res-piratory symptoms were observed. The durations from illness onset to first hospital admission, first Favipiravir treatment, and discharge up to 14 days were measured. Viral loads were considered in cycle threshold (Ct) value analyses. Analyses considered viral load for comparisons of Ct values by vaccine exposure groups group and self-reported symptoms. Ct value ≥30 corresponded to copy number threshold <106/mL or less, indicating a low viral RNA [22]. (Page 4, Line 140-146)

2.4. Serological Assays

Our COVID-19 diagnostic assay was a probe-based qualitative RT-PCR probe. Allplex™ 2019-nCoV Assay (Seegene, Seoul, South Korea) was used for SARS-CoV2 detection. The targeted COVID-19 genes detected here included nucleocapsid (N), envelope (E) of Sarbecovirus and RNA-dependent RNA polymerase (RdRp) of COVID-19 according to the manufacturer’s instructions and described previously [23]. (Page 4, Line 148-152)

The anti-SARS-CoV-2 RBD IgG assay linearly measures the level of antibody between 21.0-40,000.0 arbitrary unit (AU)/mL, which was converted later to WHO International Standard concentration as binding antibody unit per mL (BAU/mL) following the equation provided by the manufacturer (BAU/mL=0.142 x AU/mL) [24]. The level greater or equal to the cutoff value of 50 AU/mL or 7.1 BAU/mL was defined as seropositive. Surrogate Virus Neutralization Test (sVNT) was undertaken against original (Wuhan) strain and the Delta (B1.1617.2) strain due to its availability during the study period. Briefly, plasma was pre-incubated with horseradish peroxidase conjugated-receptor binding domain protein (HRP-conjugated RBD protein). Subsequently, mixture was transferred to each well containing Streptavidin bound with Biotin conjugated angiotensin-converting enzyme 2 (ACE2). The plate was washed, substrate and stop solution were added. Finally, the optical density absorbance was measured using spectrophotometer at 450 nm. Inhibition rate was calculated through this formula:

Sample diluent was used as the negative control. White blood cell count, C-reactive protein, and D-dimer results were retrieved from electronic medical record from patients who were readmitted to the hospital. (Page 4, Line 156-174)

2.5. Statistical Analysis

The NCT is closely related to clinical manifestation and disease progression in COVID-19 patients. (Page 4, Line 180-181)

Normally distributed continuous variables were summarized as the mean ± SD; otherwise, median (interquartile range, IQR) was used. Categorical variables were expressed using numbers and percentages. (Page 5, Line 184-186)

Point 3: Results

Point 3.1: Paragraph 3.1. The number of patients in each age class should be considered, as well as the incidence. In addition, these data may be due to individual susceptibility to virus variants, different vaccine type, reduced isolation measures, etc., and need to be discussed.

Point 3.2: No patients with Omicron were treated with dexamethasone (Favi/Dexa): how to compare the data with Delta patients?

Point 3.3: Tables 1, 2, and 3: uniform text and spacing characters in notes.

Point 3.4: Table 3. Write in full the meaning of the abbreviation RR.

Point 3.5: The figures are too small to display, especially fig. 2 C-L. I suggest reducing the number of figures, choosing the most significant ones for a better understanding of the results.

Point 3.6: Lines 250-252. I suggest reporting the ages in the text. Age are not shown in figure S4, so why is it cited here?

Response 3:

Response 3.1: We have addressed the comments carefully in both results and discussion section.

We have added “The proportion of COVID-19 infections was highest in the group aged 25 years or more during the Delta wave (1470 [54.4%]) and during the Omicron wave (1220, [49.3%]), whereas, increased proportion of COVID-19 infections was observed in the young during the Omicron pandemic (1015, [41%]).” (Page 5, Line 194-197)

“According to age groups, there was no differences between common symptoms in all age groups either Delta or Omicron pandemics. Whereas the Ct values in all age groups of Delta pandemic (Ct 20.1 to 21.8) were higher than that of Omicron pandemic (Ct 18.8 to 20.6) (Supplementary Table S2-S3).“ (Page 5, Line 208-212)

“In addition, in the Omicron pandemic, children made up a larger proportion of patients than in the previous infection waves. One potential explanation was that the Omicron variant’s extremely high transmissibility, when coupled with a lack of built-up immunity from yet authorized vaccination for 5-11-year old in Thailand in the early 2022 or past infection in the young children, leaved children more vulnerable to Omicron, compared with adults who had access to vaccines for months. Another, other restrictions and isolation policies were eased at the same time, and parents had the ability to return to the workplace, where transmission also occurred and immunity waned.” (Page 15, Line 416-423)

Response 3.2: We have addressed the comments in discussion section.

“No patients with the Omicron were treated with Dexamethasone (Favi/Dexa) in HI system due to reduced severity of the Omicron infection, and increased hospitalization availability for worsen cases. However, to compare both waves, we had excluded patients in Favi/dexa group in the Delta wave from the analyses and the p-value was not affected in all parameters.” (Page 16, Line 482-486)

Response 3.3: We have addressed your comments and agreed. We use font Palatino Linotype 8 pt. in size for text in the table, and 9 pt. in size for table legends/footnotes. (Page 5-9)

Response 3.4: We already described the abbreviation RR in the footnote of Table 3. (Page 9, Line 276)

Response 3.5: Thank you for your comment. We realized this issue and decided to split the figure 1 and 2 in the Results Section. 

Response 3.6: We have changed Figure S4 to Figure S3 and cited in Page 9, Line 312 and Page 13, Line 352 instead.

Point 4: Discussion

In addition to some points as previously indicated, a more in-depth discussion is needed. There has been a lot of data recorded on symptoms in respect to vaccine status as well as to age and gender, so they deserve a discussion of differences observed in respect to all of these parameters. This will help to better support conclusions.

Response 4: Thank you for the suggestion. We have addressed the reviewer’s comments.

“Data showed that primary immunization with two doses of ChAdOx1 or CoronaVac vaccine provided limited protection against symptomatic disease caused by the Delta and the Omicron variants and vaccine effectiveness waned quickly. Higher neutralizing activity was observed after a booster dose [29]. However, we are the first to show that breakthrough COVID-19 infection with prior vaccination were associated with a significantly lower number of symptoms and fatigue even in the mild-to-moderate COVID-19 disease. On the other hand, the lower number of symptoms was, the lower antibody titers were. The difference is risk between the Delta and Omicron varied considerably with age but not gender. The risk of hospitalization differed the most for those aged 60 years (50% lower for Omicron compared with Delta) [30]. In contrast, for children under the age of 12 years, there was no significant difference in risk of hospitalization between Omicron and Delta. The risk of death remained minimal in children. Our unpublished data showed that infected children during the Delta and Omicron pandemic were hospitalized due to insufficient oral intake, anorexia, gastrointestinal tract symptoms and hypovolemia rather than increased COVID-19 severity.” (Page 15, Line 400-415)

“In the mild-to-moderate COVID-19 cases during the Delta and Omicron pandemics, IgG and sVNT were higher in patients with more severe common COVID-19 symptoms. The titers were also associated with high viral loads and older individuals (who were generally vaccinated and had more severe symptoms than asymptomatic individuals) [7]. Higher igG and sVNT was showed in breakthrough COVID-19 patients vaccinated with either CoronaVac-prime or ChAdOx1-primary dose. The titers peaked around 2 to 3 months PC and remained stable for at least 3 months.” (Page 15, Line 426-430)

Point 5: Supplementary

Page 4. X” vaccination”, capital letter.

Page 9. I did not find the note “7” in the text.

Response 5:

- We have edited vaccination with capitalization. (Supplementary Appendix Page 5, Line 8 in the context under Figure S1)

- The Serological Assays Section moved to the main text. (Page 4, Reference 14)

Reviewer 2 Report

In this clinical evaluation, Lastname, F, et al investigated the clinical outcomes and laboratory data of 5181 patients with mild-to-moderate COVID-19 under home isolation. They evaluated anti-receptor binding domain immunoglobulin G (anti-RBD IgG) and surrogate viral neutralizing (sVNT) activity in 495 individuals post-COVID-19 infection during the Delta pandemic. They showed that two-dose vaccine regimen reduced hospital readmission and death. In addition, anti-RBD IgG and sVNT against Delta were higher among older individuals post-COVID-19 infection. Thus, they conclude that after a full vaccination course, breakthrough mild-to-moderate Delta and Omicron infections have limited immunogenicity. 

Overall, the manuscript was written well. However, this is a simple clinical report and there is no any mechanistic investigation involved. In addition, T cell-mediated adaptive immunity that is critical for the protection of  COVID-19 infection is ignored. Cytokines associated with COVID-19 infection are also not examined. The value of this report is modest.

Author Response

Response to Reviewer 2 Comments

Point 1: In this clinical evaluation, Lastname, F, et al investigated the clinical outcomes and laboratory data of 5181 patients with mild-to-moderate COVID-19 under home isolation. They evaluated anti-receptor binding domain immunoglobulin G (anti-RBD IgG) and surrogate viral neutralizing (sVNT) activity in 495 individuals post-COVID-19 infection during the Delta pandemic. They showed that two-dose vaccine regimen reduced hospital readmission and death. In addition, anti-RBD IgG and sVNT against Delta were higher among older individuals post-COVID-19 infection. Thus, they conclude that after a full vaccination course, breakthrough mild-to-moderate Delta and Omicron infections have limited immunogenicity.

Overall, the manuscript was written well. However, this is a simple clinical report and there is no any mechanistic investigation involved. In addition, T cell-mediated adaptive immunity that is critical for the protection of COVID-19 infection is ignored. Cytokines associated with COVID-19 infection are also not examined. The value of this report is modest.

Response 1: Thank you for your comments.

We also added that “Even though we did not have a complete set of these parameters for every subject due to the nature of retrospective data from mild-to-moderate COVID-19, we had some patients with worsening conditions who were eventually hospitalized and whose blood was examined. Our results indicated that, during the Delta but not the Omicron wave, these patients had increased neutrophils and lymphocytopenia and activation of the coagulation cascade. However, there were reports that Omicron patients had ab-normal levels of neutrophils, lymphocytes, and monocytes and demonstrated signs of coagulopathies [33-35]. Some studies investigated the T cell immunity induced after SARS-CoV-2 infection in mild symptomatic cases, showing S-SARS-CoV-2-specific IFN-γ T cell response was developed [36]. CD4+ T cell responses against SARS-CoV-2 were more prevalent than CD8+ T cell response in adults with mild-to-moderate COVID-19 infection [37]. Still, more in‐depth research on the underlying etiology is necessary.” (Page 16, Line 449- 460)

During the study period, we decided not to analyze the t cell mediated immunity and related cytokines because almost all patients with mild-to-moderate symptoms were isolated at home since COVID-19 was diagnosed. Therefore, it is impractical to ask them to have blood drawn during the infection and the Delta pandemic was over at the study period. Still, we are working on the study in patients with long COVID-19 during Omicron pandemic which we are addressing the questions you are raising. And this study will be ready to submit soon.

Reviewer 3 Report

the research is potentially interesting

Numerous changes are needed in order for the work to meet the criteria

1. Consent for data use who approved?

2. From which database was the data used? what was the approach to the base like? why is there no patient consent?

3. Respondents' rights are violated if written consent is not obtained for data collection and public disclosure without the consent of individuals?

4. On the basis of which law was the data on patients taken?

5. Publishing such works without consent is a violation of the law on patient data protection! Which entails criminal liability

6, the name imunity responce is comprehensive and does not correspond to the data presented in the paper

7. The paper mostly shows clinical symptoms and other findings that have nothing to do with immune reactions, and one is written in the title and completely different in the results

8, When we say immune response, it is a very broad term that would include many reactions of the immune system and does not refer at all to the clinical picture shown here.

Author Response

Point 1: the research is potentially interesting

Numerous changes are needed in order for the work to meet the criteria

  1. Consent for data use who approved?
  2. From which database was the data used? what was the approach to the base like? why is there no patient consent?
  3. Respondents' rights are violated if written consent is not obtained for data collection and public disclosure without the consent of individuals?
  4. On the basis of which law was the data on patients taken?
  5. Publishing such works without consent is a violation of the law on patient data protection! Which entails criminal liability

Response 1: Thank you for your comments and important ethical concerns. We really appreciated your awareness. At first, we tried to make the manuscript more concise; therefore, some of important details might be missing. We would like to respond to these comments as follow.

We have added more details.

“Someone who met inclusion criteria would be considered to have mild symptoms, or perhaps be asymptomatic, and would be referred to Siriraj-Home system (SI-Home) where medicine will be delivered by health personnel within 24 hours rather than being relegated to a field hospital or other potentially unpleasant arrangement. Data relating to clinical information, and laboratory test findings were retrieved from patients’ electronic medical records without any personal identifiable information were retrieved after IRB approval.”(Page 3, Line 114-120)

“All participants provided informed consent for this study.” (Page 3, Line 125)

This is mainly a retrospective data with a subset of 495 participants who provided consent for a follow up study. Therefore, a majority of our data set was retrieved from an electronic medical record which we have received from the hospital IT department. However, we did not link any identifying information from patients nor does we reveal any personal data. And it is almost impossible to ask them to sign a consent again due to the nature of study. We have added an approval COA letter (COA no. Si 732/2021 entitled “Treatment outcomes, cost-effectiveness analysis, feasibility and acceptability, and serology testing of COVID-19 patients in ambulatory setting” and COA no.  Si833/2021 entitled “Development of SARS-CoV-2 neutralizing antibody test kit and treatment outcome and longitudinal study in COVID-19 patients with home isolation setting.”

We follow the international standard guidelines strictly in accordance with Mahidol University Act 2007, Section 24(2). And this institutional Review Board (IRB) certified as a standard in the Human Research Protection Program from the United States organization, AAHRPP or the Association for the Accreditation of Human Research Protection Programs since December 2017. And the IRB receives regular auditing and evaluation annually and have been recognized on the quality of the committee by FERCAP in 2009, 2012, 2016 and 2020.

Also, IRB approved a waiver of informed consent for the retrospective medical record review since we followed all of the following criteria based on 45 CFR § 46.116 (d).

  1. The research involves no more than minimal risk to the subjects;
  2. The waiver or alteration will not adversely affect the rights and welfare of the subjects;
  3. The research could not practicably be carried out without the waiver or alteration; and
  4. Whenever appropriate, the subjects will be provided with additional pertinent information after participation.

We also added the informed consent letter that we provided to participants with some translation to address your important concern.

Point 2:

  1. The name immunity response is comprehensive and does not correspond to the data presented in the paper
  2. The paper mostly shows clinical symptoms and other findings that have nothing to do with immune reactions, and one is written in the title and completely different in the results
  3. When we say immune response, it is a very broad term that would include many reactions of the immune system and does not refer at all to the clinical picture shown here.

Response 2: Thank you for your comment. I agreed with the editor and the reviewer’s comment. I also would like to propose three titles as follow.

  1. Immune response against mild-to-moderate breakthrough COVID-19
  2. SARS-CoV-2 antibody response against mild-to-moderate breakthrough COVID-19 in home isolation setting in Thailand
  3. Immune response against mild-to-moderate breakthrough COVID-19 during Delta and Omicron pandemic

Round 2

Reviewer 1 Report

The manuscript was revised according to the suggestions, and in the present formi it has greatly improved. Only a few revisions remain prior the publication, as follows.

·        Add the proper references for Remdesivir and Monipiravir, at lines 84-85.

·        As reported by Aranha (23), RT-PCR was performed on nasopharyngeal swab specimens from patients, so this assay should be reported in a paragraph with a different title (not “Serological Assays”).

·        In the legend of Table 3, write “RR, relative risk” before than “aRR, adjusted relative risk”.

·        I cannot find the reference to figure 6 in the text, please insert.

Author Response

The manuscript was revised according to the suggestions, and in the present formi it has greatly improved. Only a few revisions remain prior the publication, as follows. 

Point 1: Add the proper references for Remdesivir and Monipiravir, at lines 84-85. 

Response 1: Thank you for your comments. We also added that “Remdesivir [13], a monophosphoramidate prodrug of the nucleoside GS-441524, is recommended only for severe disease due to limited access. In addition, monipiravir [13], the oral prodrug of beta-D-N4-hydroxycytidine (NHC) and anti-SARS-CoV-2 monoclonal antibodies were not available during the study period.” (Page 2, line 84-88)

Point 2: As reported by Aranha (23), RT-PCR was performed on nasopharyngeal swab specimens from patients, so this assay should be reported in a paragraph with a different title (not “Serological Assays”).

Response 2: We have added details about diagnosis of COVID-19 in the highlighted sentences in Materials and Methods Section. 

2.4. Diagnosis of COVID-19  

Diagnosis of COVID-19 is made based on the detection of ≥2 SARS-CoV-2 genes by RT-PCR from nasopharyngeal (NP) swab, throat swab, and/or any respiratory samples as previously described [24]. Our COVID-19 diagnostic assay was a probe-based qualitative RT-PCR probe. Allplex™ 2019-nCoV Assay (Seegene, Seoul, South Korea) was used for SARS-CoV2 detection. The targeted COVID-19 genes detected here included nucleocapsid (N), envelope (E) of Sarbecovirus and RNA-dependent RNA polymerase (RdRp) of COVID-19 according to the manufacturer’s instructions and described previously [25]. (Page 4, Line 149-158) 

Point 3: In the legend of Table 3, write “RR, relative risk” before than “aRR, adjusted relative risk”.

Response 3: Thank you for your suggestion. We have changed “RR, relative risk” before “aRR, adjusted relative risk”. (Page 9, Line 288-289)

Point 4: I cannot find the reference to figure 6 in the text, please insert.

Response 4: Thank you for your comment. We have changed “Figure 6A-6D” to “Figure 6. (Page 13, Line 375-376)

Reviewer 2 Report

The authors fairly addressed my main concern and could not perform analyses in the T cell-mediated responses and cytokine production.

It can be a case report, but has no much scientific value in the area.

Author Response

Point 1: The authors fairly addressed my main concern and could not perform analyses in the T cell-mediated responses and cytokine production.

It can be a case report, but has no much scientific value in the area

Response 1: We have addressed the comments carefully in discussion section. 

We have added “The adaptive immune response was a key element of the clinical outcome after SARS-CoV-2 infection and supported vaccine efficacy. T cell responses activated early and correlated with protection but were relatively weakened in severe COVID-19 and are associated with intense activation and lymphopenia [38]. Inflammatory cytokines such as IL-6, IL-8, IL-1β, TNF-α, IFNƔ-induced protein10 (IP-10), granulocyte-macrophage colony-stimulating factor (GM-CSF), and chemokines (CC motif) ligand 2 (CCL2), CCL-5, and CCL3 were generally produced by macrophages, mast cells, endothelial and epithelial cells during the innate immune response. Many studies have shown that elevated IL-6 significantly affected the onset of cytokine storm [39,40]. IL-6 played a pleiotropic role in the immune system and was crucial for the formation of TH17 and follicular helper T cells. However, IL-6 could block cytotoxic CD8 + T cells by inhibiting IFN-Ɣ secretion. In addition, IL-6 could impair the cell-induced antiviral response in the cytokine storm. Our unpublished data showed some significant IL-6 level in some covid-19 patients. However, we need to clarify the factors predisposing to cytokine storm and the other Inflammatory cytokines.” (Page 16, Line 467-481)

“Gao Y et al. [43] demonstrated that SARS-CoV-2 spike-specific CD4+ and CD8+ T cells elicited by BNT162B2 vaccination or previous infection remain largely intact against the Omicron variant. Together with intrinsic viral factors, these immune reactivities, in part, explain why severe disease appeared to be minimal after breakthrough infection with this particular variant.” (Page 16, Line 485-490)

Reviewer 3 Report

accept

Author Response

Thank you very much.